# Motives and Passion of Adults from Pakistan toward Physical Activity

**DOI:** 10.3390/ijerph19063298

**Published:** 2022-03-11

**Authors:** Muhammad Badar Habib, Selina Khoo, Tony Morris

**Affiliations:** 1Centre for Sport and Exercise Sciences, Universiti Malaya, Kuala Lumpur 50603, Malaysia; badar.habib@siswa.um.edu.my; 2Department of Physical Education & Sports Sciences, The Islamia University of Bahawalpur, Bahawalpur 63100, Pakistan; 3Institute for Health & Sport, Victoria University, Melbourne 6012, Australia; tony.morris@vu.edu.au

**Keywords:** motives, passion, physical activity, physical inactivity, adults

## Abstract

Globally, a large proportion of people do not participate in adequate physical activity to gain health benefits. Physical inactivity is a primary risk factor for global mortality. Research suggests that motives and passion influence participation in physical activity. The main purposes of the present study were to examine the relationship between motives and passion for participation in physical activity among adults, and to examine whether motives and passion predicted physical activity. Prior to data collection, we translated and validated the Physical Activity and Leisure Motivation Scale (PALMS) and the Passion Scale (PS) into Urdu. With the research sample, both translated questionnaires showed acceptable internal consistency, test-retest reliability, and factorial validity. To address the main purposes, we then employed a quantitative, cross-sectional research design, with a sample of 572 adults between 18 and 65 years (31.51 ± 10.25), who regularly participated in moderate-to-vigorous physical activity. Participants completed the PALMS, the PS, and the International Physical Activity Questionnaire (IPAQ). Correlation coefficients showed strong relationships between motives and harmonious passion, and weaker, negative relationships between motives and obsessive passion. We used stepwise multiple regression to test whether motives and passion subscales were significant predictors of overall PA measured by the IPAQ. In step 1 of the regression model, mastery and physical condition motives were significant predictors of overall physical activity. Further, in step 2, the mastery motive and harmonious passion were significant predictors of overall physical activity, accounting for 26% of the variance, with *F* (5, 566) = 38.84, *p* < 0.01. The present study provides a basis for physical activity interventions examining whether increasing motives and passion leads to higher adherence to and lowered dropout from physical activity.

## 1. Introduction

Despite acknowledgement of the benefits of physical activity (PA) for physical health and mental wellbeing, worldwide, levels of PA remain stubbornly low, even decreasing over the 15 years between 2001 and 2016 [1]. One approach to increase PA participation focuses on psychological factors. Early studies have related motivation [2] and passion [3], independently, to increases in PA. Theoretically, in self-determination theory (SDT), Deci and Ryan [4] proposed that motives for PA and passion for PA are closely related. Thus, in the present study, we aimed to examine whether motives for PA and passion for PA together have a stronger influence on PA than each does alone. A secondary aim was to examine whether motives for PA and passion for PA are closely related, as predicted by SDT.

PA has been linked with physical, psychological, and social benefits, including reduced prevalence of cardiovascular disorders, obesity, diabetes, cancer, social anxiety and depression [5,6,7]. These benefits are experienced by individuals from different age groups, gender, and occupations [8]. The literature suggests that doing PA for a longer time reduces stress, anxiety, and depression among adults [9].

In adults and older adults, PA has been associated with prolonged, healthier lives [10]. Despite its well-known and established benefits, a large proportion of the population does not undertake adequate PA to experience sustained and healthier lives [11]. The World Health Organization (WHO) recommends that adults aged 18–64 years engage in at least 150 min of moderate-intensity PA, 75 min of vigorous-intensity PA, or an equivalent of moderate- and vigorous-intensity PA, per week to achieve health benefits [12]. However, even a minimum of 15 min of PA per day can decrease mortality by 14%, potentially adding three years to life [13].

Physical inactivity is the fourth leading risk factor for global mortality and is estimated to cause 6% of deaths worldwide [14]. It has led to increased risks of cardiovascular diseases, obesity, diabetes, hypertension, stroke, and associated physiological disorders [15,16]. This increasing trend in physical inactivity has not only resulted in an increase in disease prevalence, but also poses a substantial economic burden globally [17]. In order to reduce this health problem, the WHO has launched a global action plan to reduce physical inactivity by 10% by 2025, and 15% by 2030 [12].

There is ongoing research to discover how to encourage greater participation in PA across the world’s population [18,19]. Motivation is a key factor that influences PA [20]. It is one of the pivotal factors that affects not only initiation, but also sustainability and maintenance of PA [21]. Motivation refers to a general psychological process, about which there are many different theories and models, and, consequently, a vast and diverse literature. Motives are specific constructs that underlie the reasons people give for their behaviour. Motives for PA have been studied in sports and exercise related to physical, psychological, and social aspects [22]. Understanding the relationship between motives and PA has great potential for promoting participation in PA for health and wellbeing [23].

The theoretical approach most strongly associated with linking motivation to behaviour, including PA, is SDT. SDT, a wide-ranging and grounded macro-theory, is based on the perspectives of cognitive and humanistic psychology [4,24,25]. SDT is an influential approach for understanding human motivation. According to Deci and Ryan [26], intrinsic motivation is the doing of an activity based on individuals’ internal satisfaction (joy, fun, and pleasure), whereas extrinsic motivation refers to doing an activity for instrumental reasons, as a means to seek some objectives (such as money, awards, status, or fitness) [27]. In sport and PA, researchers have examined motives based on age, gender, and type of activity [2,28,29]. Researchers have reported that motivation for PA changes with age [2]. Older adults were more extrinsically motivated to participate in PA, compared to younger adults [2]. Molanorouzi et al. [2] also found that males were more motivated by mastery and competition/ego and females were more motivated by appearance and physical condition. Research on age and gender differences in motives for participation in PA, further indicates that adult females and males show different motives for PA participation [30]. According to Egli et al. [30], most of the male adults they studied were intrinsically motivated, whereas a larger proportion of females were extrinsically motivated. Molanorouzi et al. [2,30] found differences in motives for participation in PA that were related to the type of PA, including that racquet sport players were most motivated by mastery and competition, whereas non-competitive exercisers were most motivated by psychological and physical condition and appearance.

Reliable and valid instruments are very important to identify and measure adults’ motives for participation in PA. Several questionnaires have been developed for this purpose. Notable theoretically driven measures are the Motivation for Physical Activity Measure (MPAM) [31], which includes three motives derived from self-determination theory (SDT), and the Motivation for Physical Activity Measure—Revised (MPAM-R) [32], which includes five motives. Empirically derived measures, such as the Participation Motivation Questionnaire, (PMQ) [33] identified more motives for PA participation by asking participants their reasons for participating in PA. However, the PMQ was never formally validated. The Physical Activity and Leisure Motivation Scale (PALMS) [34] is a reliable and valid questionnaire for examining motives for participation in PA, that is based both on the earlier atheoretical PMQ, and on the theoretically based MPAM-R. Use of PALMS is supported by its strong validity. PALMS has been validated in English [35], Hebrew [36], and Malay [37]. Hence, PALMS is a useful measure of motives for participation in PA, which is being used by researchers around the world to investigate the participation motives that individuals use to become engaged in physical activities [38].

This research suggests that studies should be conducted to identify which motives for participation in PA have a noteworthy influence on the amount of PA people do. These motives can be used to change the increasing trend of sedentary behaviour among adults. Intrinsic motives, such as mastery, satisfaction, and enjoyment are proposed to be the main drivers of long-term participation in PA [39]. It has been observed that people with higher ratings on intrinsic motives had increased benefits in activity and function [2,40].

Passion is defined as a strong inclination towards a personally meaningful and valued activity that individuals love, and on which they invest substantial time and energy [41]. Passion seems to ignite motivation, well-being, and energetic involvement in a specific task, be it physical or mental. According to Curran et al. [42], passion is “a special relationship with an activity that one loves” (p. 47). Passion is considered to be a strong motivational energy force towards anything that is important, liked, and in which a significant amount of time is invested [43]. Passion is internalized into a person’s identity when the individual enjoys the activity [44]. Research indicates that passion can be directed towards PA [3]. Vallerand proposed the Dualistic Model of Passion (DMP), depicting two types of passion, namely harmonious passion (HP) and obsessive passion (OP) [41,45]. Vallerand et al. [41] developed the Passion Scale (PS) to measure the two types of passion. The PS consists of 17-items. Six items form one subscale that measures HP, and a further six items form a second subscale measuring OP. The remaining five items measure individuals’ passion criteria [41]. The PS has been validated in English, and translated into several other languages [46,47,48,49]. In the DMP, it is proposed that both types of passion have important roles in determining the participation of individuals in PA [43]. HP is related to positive, affective outcomes that are adaptive, whereas OP is related to negative or maladaptive outcomes [50,51]. OP refers to the passion in which pressure (physical, mental, social, and psychological) tends to force individuals to engage in some activity or work, with intense commitment, but with little or no enthusiasm and motivation [41]. It is proposed that withdrawal from the activity causes high levels of stress [10]. Each type of passion results in different outcomes. HP results in positive, adaptive, and effective outcomes, whereas OP has negative outcomes that are maladaptive and unsuccessful for individuals [42,45,52]. Motivation and passion are closely linked [53] and they both appear to affect participation in PA [54]. Thus, it is important for researchers to examine how motives for participation in PA and HP and OP are related to levels of PA participation, both separately and together. This research is significant because promoting PA is a global issue for health and wellbeing. Theory and early research suggest that enhancing motives and passion for PA could have a powerful influence on PA participation worldwide.

Aspects of SDT [4] suggest that motives and passion are closely connected. In the SDT sub-theory organismic integration theory, Deci and Ryan [4] proposed a motivation integration continuum running from amotivation, the absence of motivation, through several levels of extrinsic integration, to levels of progressively more intrinsically motivated action, ending with full intrinsic motivation, which is proposed to be the goal for any activity. However, Vallerand et al. [41] proposed that passion should be conceptualized as an intense form of motivation, distinguished from intrinsic motivation because behaviour based on passion is internalized, whereas intrinsic motivation is not. This proposed close association between intrinsic motivation and passion suggests that a combination of intrinsic motivation and passion should provide a very strong basis for long-term participation in any activity. As far as we are aware, there is no published research that has addressed this proposition. Thus, we believe that this is the first study in the field of sport and exercise psychology to examine whether motives for participation in PA and passion are related, and whether they influence amount of PA, either independently or in combination. The significance of this study is that, if motives for participation in PA and passion for PA together have a stronger association with amount of PA participation than they do separately, this could have a significant impact on interventions to promote PA globally for population health and wellbeing.

Pakistan is a country that has serious concerns with levels of inactivity. A total of 81% of Pakistanis do not participate in adequate PA to experience health benefits [55]. According to research on reasons for participating in PA, lack of motivation, few fitness clubs, and little family support are amongst the main barriers to PA in Pakistan [56]. Research is needed to identify ways to increase participation in PA in Pakistan. Motives and passion are two influential psychological variables that could enhance participation in PA in the Pakistani context.

Thus, the major focus of the present study is to examine the influence of motives and passion on levels of participation in PA among adults in Pakistan. The aims of the study are, first, to examine whether motives for participation in PA are related to HP and OP; second, to examine whether motives for participation or HP or OP separately predict the extent of participation in PA; and third, to examine whether motives for participation in PA and HP and OP combined predict a larger percentage of the variance in PA than motives or passion independently.

## 2. Materials and Methods

In this study, we employed a cross-sectional research design to examine the relationship between motives for participation in PA and harmonious and obsessive passion, and the effect of motives for participation and harmonious and obsessive passion on participation in PA. The study was approved by the University of Malaya Research Ethics Committee (UM. TNC2/UMREC-373). Participation in the study was voluntary and all participants provided written informed consent.

We recruited all participants through convenience sampling. We invited Pakistani adults aged between 18 and 65 years, who regularly participated in low-intensity to vigorous-intensity PA, and took part in various sports and non-competitive PA, to participate in the study. We recruited them from sports clubs, fitness centres, and recreational parks in Pakistan. We informed them about the objectives of the study and their right to withdraw from the study at any time. We only used demographic information related to gender, age, experience in PA, and type of sports and PA of the participants for the study analysis. We used codes to identify sets of questionnaires, so participants were not personally identified.

### 2.1. Translation

Prior to data collection, we translated the PALMS and the PS from the source language (English) to the target language (Urdu), using back-translation based on Brislin’s model [57]. We conducted a pilot study with 25 participants. We calculated internal consistency reliability, which met the threshold criteria for the PALMS and the PS in Urdu, which was 0.7 [58], so we considered that the questionnaires were ready for use in the main study [59].

### 2.2. Participants

A total of 572 Pakistani adults (393 males, 179 females) participated in the study. They were aged between 18 and 65 (31.51 ± 10.25) years and took part in various types of PA, comprising sports and non-competitive PA, including yoga, gym exercise, racquet sports, team sports, swimming, jogging, walking, and strength exercise. We calculated the sample size based on the a priori Sample Size Calculator for correlation and multiple regression from an online statistical calculator (www.danielsoper.com accessed on 10 December 2021). Based on the anticipated effect size (*f*^2^ = 0.03), statistical power level (0.80 or 80%), number of predictors (8), and probability level (0.05), we calculated the minimum required sample size for this study to be 506 [60]. Oversampling by 15% (*n* = 76) was undertaken, to allow for attrition, resulting in a sample size of 582.

### 2.3. Measures

#### 2.3.1. Demographic Information Form

We asked participants to provide demographic information related to gender, age, academic qualifications, experience in PA, type of sports and PA in which they participated at the time of the study, and number of days of the week in which they engaged in PA.

#### 2.3.2. Physical Activity and Leisure Motivation Scale (PALMS; Morris & Rogers, 2004)

The PALMS consists of 40 items and assesses eight motives for participation in PA (mastery, enjoyment, psychological condition, physical condition, appearance, others’ expectations, affiliation, and competition/ego). Responses to PALMS are made on a 5-point Likert scale ranging from 1 (strongly disagree) to 5 (strongly agree). Thus, scores for each sub-scale range from 5 to 25; a higher score indicates higher motivation for participation in PA. The PALMS is a reliable (Cronbach’s alpha α = 0.70–0.92) and valid measure of motives for participation in PA [2,37] that has been validated in a number of languages [2,36,37] and used widely in research [40,61].

#### 2.3.3. Passion Scale (PS; Vallerand et al., 2003)

The PS consists of two six-item subscales for assessing HP and OP towards an activity and a 5-item sub-scale to identify individual passion criteria, assessed on a 7-point Likert-type scale ranging from 1 (not agree at all) to 7 (very strongly agree). Scores range from 6 to 42 with a higher score indicating more passionate characteristics. The PS is a highly reliable (Cronbach’s alpha α = 0.75–0.90) and valid measure [41,62].

#### 2.3.4. International Physical Activity Questionnaire (IPAQ; Craig et al., 2003)

We used the 7-item short form of IPAQ to assess the PA level of participants based on the number of days and minutes per day of doing PA in the past seven days [63]. IPAQ assesses three types of PA, namely walking, moderate intensity, and vigorous intensity activity. The amount of PA was converted into estimated metabolic equivalents (METs) and calculated into MET-minutes/week. IPAQ showed good reliability and validity with test-retest reliability of Spearman’s rho, = 0.66 to 0.88 (averaging 0.76). IPAQ is a valid measure of energy expenditure by walking, moderate and vigorous intensity activity, and sitting time [64,65]. IPAQ is available in 21 languages, including Urdu, which is the national language of Pakistan and the first language of all participants in this study.

### 2.4. Procedure

We recruited participants from sports clubs, fitness centres, and recreational parks in five provinces in Pakistan (Balochistan, Gilgit-Baltistan, Khyber-Pakhtunkhwa, Punjab, and Sindh). We collected data from November 2018 to January 2019. We briefed participants about the study and distributed information sheets. We encouraged participants to ask questions to clarify the procedure and we answered the questions. Participants who then agreed to participate completed the consent form, demographic information sheet, PALMS, PS, and IPAQ. A total of 582 questionnaires were distributed. Ten questionnaires were discarded because of missing items, leaving 572 responses for analysis.

### 2.5. Statistical Analysis

We conducted statistical analyses using the Statistical Package for Social Sciences (SPSS for Windows, Version 24, SPSS Inc. Chicago, IL, USA) and AMOS version 24. Confirmatory factor analysis (CFA) was conducted to test the validity of the translated PALMS and PS. We calculated descriptive statistics (means, SDs, SEs and frequencies) for demographic attributes of the study, and scales and subscales of PALMS, PS, and IPAQ questionnaires. To examine whether motives for participation in PA were related to HP or OP, we calculated Pearson’s product moment correlations between the eight participation motives in the PALMS and HP and OP. To examine whether motives for participation or HP and OP separately predicted the extent of participation in PA, we conducted hierarchical regression analysis. To determine which motives to enter into the regression analysis, we chose a threshold correlation of *r* = 0.37. This process was applied because including all 8 motives would have required a large number of cells, resulting in a very large total sample size or reduced power. Excluding motives with lower correlations increases the power of the analysis, without great risk of missing significant variables. In such circumstances a correlation between *r* = 0.35 and 0.4 is usually selected. We chose a relatively conservative threshold of *r* = 0.37. In the first step, we entered participation motives that had correlations above the predetermined threshold of, *r* = 0.37, as predictor variables and IPAQ METs for overall PA was the criterion variable. We identified motives that predicted a significant percentage of the variance. In the second step, we added HP and OP to the regression analysis as predictor variables with the same criterion variable. We examined the additional variance accounted for by HP and OP. We identified passion scales that predicted a significant percentage of the variance. To test whether motives for participation in PA and HP or OP combined predicted a larger percentage of the variance in PA than motives or passion independently, we examined the total variance predicted by the PALMS subscales chosen for this analysis and the two PS subscales. In this way, we identified motives and passion subscales that together predicted a significant percentage of the variance.

## 3. Results

### 3.1. Validation of Questionnaires

PALMS and the PS were examined for test-retest reliability and internal consistency reliability. Then, PALMS and the PS were validated through CFA (*N* = 572). For PALMS, test-retest reliability was examined based on Pearson’s Product–moment correlations in a subsample of 83 participants over a 4-month period. Correlations for all eight motives were greater than *r* = 0.83. Internal consistency for the eight 5-item motive subscales was all well above the standard threshold of 0.7 [58], with the lowest Cronbach’s alpha coefficient (α) being α = 0.84. In addition, the Composite Reliability index (CR) was greater than 0.7 [66] for all eight motives and the Average Variance extracted (AVE) was larger than the agreed threshold of 0.5 [67] for all eight motives. Based on modification indices [68,69], CFA revealed CFI and NNFI values to be 0.93 and 0.92, respectively, while χ^2^/df was 2.59. RMSEA and SRMR for the modified first-order model were 0.05 and 0.04, respectively. Thus, the data fit the model and the Urdu language version of PALMS was considered to be suitable for use in the main study (See the Appendix A for details of the validation process).

For the PS, test-retest reliability was also examined based on Pearson’s Product-moment correlations in the subsample of 83 participants over a 4-month period. The correlation for HP was *r* = 0.88, and for OP it was *r* = 0.76. Internal consistency for the 6-item HP subscale was, α = 0.90, and the coefficient for the 6-item OP subscale was, α = 0.89, both well above the threshold of 0.7 [58]. In addition, CR was greater than 0.7 [66] for HP and OP and AVE was larger than the agreed threshold of 0.5 [67] for both passion subscales. Based on modification indices [68,69], CFA revealed CFI and NNFI values to be 0.99 and 0.98, respectively, while χ^2^/df was 2.55. RMSEA and SRMR for the modified first-order model were 0.05 and 0.04, respectively. Thus, the data were a good fit for the model and the Urdu language version of PS was suitable for use in the main study (See Appendix A for details of the validation process).

### 3.2. Correlation and Multiple Regression Analysis

The data was analysed in four key stages. In the first stage, the frequency distribution of the demographic variables was measured. In the second stage, descriptive statistics were computed for scales and subscales of the PALMS, the PS, and the PA questionnaire. In the third stage, Pearson product moment correlation coefficients (*r*) were calculated to show the relationships between motive and passion subscales, and to examine the relationships between motive subscales, passion subscales, and IPAQ walking, moderate intensity activity, and vigorous intensity activity scores. In the fourth stage, hierarchical regression analysis was performed to test whether the eight motives and two passion subscales predicted the extent of participation in PA reflected by IPAQ scores.

#### 3.2.1. Demographic Characteristics

A sample of 572 volunteers (male 68.7%, *n* = 393, and female 31.3%, *n* = 179) aged between 18 and 65 years (M ± SD; 31.51 ± 10.25) participated in the study. IPAQ scores revealed that more than half the respondents (*n* = 333, 58.2%) were classified as minimally active whereas the others (*n* = 239, 41.8%) were classified as active. The results of independent sample *t*-tests indicated significant gender differences in vigorous intensity activity, moderate intensity activity, walking, and total PA. The mean score of vigorous activity was significantly higher (*t* = 7.67, *p* = 0.01) among male respondents (M ± SD; 624.02 + 529.20) than female respondents (M ± SD; 298.77 + 303.66). The mean score of moderate activity (M ± SD; 766.16 + 385.75) was significantly higher (*t* = 6.98, *p* < 0.01) among male respondents than female respondents (M ± SD; 540.78 + 287.72). Mean walking activity was significantly higher (*t* = 4.44, *p* < 0.01) among male respondents (M + SD 1700.72 + 700.71) than female respondents (M ± SD; 1423.33 + 678.90). Mean total PA was significantly higher (*t* = 8.50, *p* < 0.01) among male respondents (M ± SD; 3090.90 + 1139.30) than female respondents (M ± SD; 2262.89 + 939.92). The results of bivariate correlation revealed significant negative correlations between age (in years) of respondents and vigorous intensity activity, moderate intensity activity, walking, and total PA scores.

#### 3.2.2. Correlations among Physical Activity and Leisure Motivation Scale, Passion Scale and International Physical Activity Questionnaire Subscales

To examine the first aim, namely determining whether motives for participation in PA were related to HP and OP, Table 1 shows the correlations among the eight PALMS motive subscales and the two PS subscales, HP and OP. Correlations between the motive and passion subscales and the IPAQ PA subscales, vigorous intensity activity, moderate intensity activity, walking, and sitting time are also included in Table 1. As can be seen in Table 1, correlations are often significant at low *r* values in large samples. In this case, every correlation in the table is significant at *p* < 0.05, right down to the correlation between HP and sitting time at *r* = 0.10. To interpret correlations in such circumstances, a threshold *r* value is normally used to identify meaningful correlations [70]. In this study, *r* = 0.35 was chosen as the minimum value for meaningful relationships [70]. All PALMS motives were correlated with HP at values well above 0.35, whereas no motives correlated with OP above 0.35. Interestingly, all correlations between motives and HP were positive, whereas all correlations between motives and OP were negative. Correlations of motives with IPAQ subscales also showed an interesting trend. No motives showed meaningful correlations with vigorous PA, only mastery showed a correlation that reached the threshold with moderate PA, and both mastery and enjoyment reached the threshold for walking. Further, correlations for all motives increased from vigorous PA to moderate PA to walking. In addition, all motives showed negative correlations with sitting time. Finally, all motives except others’ expectations showed correlations above the threshold with overall PA (which is vigorous intensity activity + moderate intensity activity + walking). HP showed a corresponding pattern to the motives. It was positively correlated with vigorous intensity activity, moderate intensity activity, and walking, but only exceeded the threshold for walking. However, it was positively correlated with overall PA at well above the threshold. HP was negatively associated with sitting time, but below the threshold. OP was negatively correlated with vigorous PA, moderate PA, walking, and overall PA, but all below the threshold. OP also showed a small, but positive, correlation with sitting time, well below the threshold.

#### 3.2.3. Multiple Hierarchical Regression Analysis

To examine whether motives for participation or harmonious and obsessive passion separately predict the extent of participation in PA and whether motives for participation in PA and harmonious and obsessive passion combined predict a larger percentage of the variance in PA than motives or passion independently, Table 2 shows results of the stepwise multiple regression analysis between motives and passion subscales as predictor variables and overall PA from the IPAQ, as the criterion variable. Because four motives (mastery, physical condition, appearance, and competition) exceeded the correlation criterion of *r* = 0.37, only these motives were entered in Step 1, and four motives (mastery, physical condition, appearance, and competition) and one Passion subscale (HP) were entered in Step 2 for the same reason. In Step 1, the model explains 23% of the variance in overall PA, with *F* (4, 567) = 42.54, *p* < 0.01. In this model, mastery and physical condition were found to be significant predictors of overall PA. In Step 2, the model explains 26% of the variance in overall PA, with *F* (4, 567) = 38.84, *p* < 0.01. In this model, mastery and HP were found to be significant predictors of overall PA, accounting for 3% more variance than the motives alone. Analyses for vigorous intensity activity, moderate intensity activity, and walking separately showed similar patterns, but the variances accounted for in Step 1 and Step 2 were largest for overall PA.

## 4. Discussion

The purpose of the present study was to examine the relationship between motives and passion for participation in PA among adults in Pakistan. First, we examined whether there was a relationship between motives and passion for participation in PA among adults. Second, we examined whether motives and passion independently predicted participation in PA among adults. Third, we examined whether motives for participation in PA and passion combined predicted PA more accurately than either motives or passion did independently. Here, we briefly comment on the validation of the PALMS and the PS in Urdu. Then, we focus on discussion of the three main elements of the purpose.

### 4.1. Validation of PALMS and PS

Results showed that the Urdu versions of PALMS and the PS attained an acceptable level of internal consistency reliability among the adult Pakistani sample, based on sound Cronbach’s α values for all eight PALMS motives, and both HP and OP for the PS. The α values for PALMS were comparable to those reported in other studies in Malaysia in English [35,37]. Results showed acceptable internal consistency reliability for the Urdu version of the PS, based on α coefficient values comparable to those reported in previous studies [48,71]. Based on the results of the separate CFAs for the PALMS and the PS in Urdu, it is clear that both PALMS and the PS reflected acceptable factor structure in Urdu, allowing the measurement of the eight motives for participation in PA and the two passion variables among adults in Pakistan. Both PALMS and the PS also showed strong test-retest reliability over a 4-month interval, a longer interval than is often used in examining test-retest reliability.

### 4.2. Relationship between Motives and Passion for Participation in Physical Activity

The first main purpose of this research was to examine the relationship between motives for participation in PA and HP and OP in the context of PA among adults. All eight PALMS motives were significantly positively correlated with HP, whereas all motives were significantly negatively, but less strongly, correlated with OP. All PALMS motives were correlated with HP at values well above the designated threshold of *r* = 0.35, whereas no motives correlated with OP above *r* = 0.35 [72]. Additionally, none of the correlations among the motives were above *r* = 0.85, which indicates that they should all be considered to measure independent motives [73]. HP showed moderate to high positive correlations with all motives, except others’ expectations, but no correlations exceeded *r* = 0.85. This indicates that there is a strong association between motives and HP, but again, HP and motives should be considered to be independent constructs. OP was less closely related to the eight motives for participation, and the relationship between OP and motives was negative, reflecting the often-undesirable nature of OP in contrast to the positive character of high levels of motives for participation.

The relationships between motives for participation and HP and OP largely supported the predictions we made, based on SDT. Primarily, we explored the prediction that passion extends the intrinsic motivation extreme of the autonomy and self-determination continuum proposed in organismic integration theory [25], as an extreme form of drive to participate in a preferred form of PA. The strong, positive relationships between motives for participation and HP provide support for this proposition. This is particularly true for correlations between HP and the two intrinsic motives, mastery and enjoyment. However, it could be argued that the correlations between the extrinsic motives and HP were of similar strength, whereas it might have been expected that they would be significant, but lower than the intrinsic motives, based on organismic integration theory. Such a claim would be supported by the observation that a primary characteristic of HP is the proposed autonomous nature of this form of passion, an aspect of HP that hinges on individuals controlling the passion, rather than passion controlling individuals, which is integral to OP [41]. Autonomous control is also a core characteristic of intrinsic motivation. In SDT, Deci and Ryan [24] highlighted satisfaction of the need for autonomy as a key psychological foundation of intrinsic motivation. We are not aware of any previous published sport and exercise psychology research that has demonstrated a strong association between motives for participation in PA and passion for PA. This is an original finding that warrants further examination. The correlations found in the present study provide support for the proposition that motives for participation in PA and HP for PA are closely associated. However, these results do not provide evidence that motives and HP work together to produce a larger amount of participation than either motives alone or HP on its own. Further light is now thrown on this claim in the exploration of the relationships between motives, passion, and PA.

### 4.3. Prediction of Physical Activity from Motives and Passion

The second purpose of the present research was to examine how the relationship between motives and passion affects participation in PA, based on the predictions that motives and passion, specifically HP, are independently associated with amount of PA. The third purpose was to examine the prediction that motives and HP together have a greater association with PA than either motives or HP alone. To examine these two predictions, we entered motives and passion subscales that were correlated with PA above the threshold of *r* = 0.37 separately into a stepwise multiple regression analysis between motives and passion subscales as predictor variables, and overall PA from the IPAQ as the criterion variable. Motives were entered in Step 1 of the regression analysis, and HP was entered in Step 2. The Step 1 analysis indicated that the motives of mastery and physical condition, predicted PA, as shown in Table 2. In this table, the percentage of PA (*R^2^*) predicted by the motives in Step 1 of the hierarchical regression analyses was 23%. This is a noteworthy proportion of the variance in PA. When HP was added in Step 2 of the analysis, the percentage of PA predicted was 26%, suggesting that HP added an additional 3% of independent variance to the analysis. Together the motives for mastery and HP accounted for a larger amount of the variance in PA than either did alone.

According to the tenets adopted within SDT [24], specific motives will predict the amount of PA that people do [74,75]. This is because people undertake the behaviours that they are motivated to do [4,76,77]. Thus, motives on which individuals score higher are associated with more of the behaviour related to those motives. SDT predicts that high levels of motives for PA behaviour will result in participation in increased amounts of PA and/or a greater intensity of PA. There is research in other countries to support this prediction. In particular, intrinsic motives, mastery, and enjoyment have been linked with increased number of hours of PA participation per week [2,78]. HP has also been related to PA [3]. Studies by Abdullah et al. [40], Molanorouzi et al. [2], and Roy [3] were conducted in Malaysia whereas the study by Ingledew and Markland [78] was conducted in England. However, these studies either tested the association between motives for participation in PA and amount of participation in PA alone, or examined the relationship between HP and OP for PA and amount of participation in PA alone. No study of which we are aware has examined the association between motives and HP together and PA participation, or tested whether they account for more of the variance together than either does alone. Along with the examination of correlations between motives for participation in PA and HP and OP for PA, this is original research that has no direct comparison in previous studies, to our knowledge. The significance of this research is that it suggests that by promoting motives, particularly intrinsic motives, and HP it should be possible to have more powerful and long-lasting effects on the amount of PA participation that people do than by influencing these variables separately. This is a proposition that warrants examination

### 4.4. Methodological Considerations and Limitations

It is important to recognize that the present study has limitations that should be considered when interpreting the findings, as well as methodological issues that might be useful for researchers to consider. First, both the motives and passion for participation in PA and the amount of PA participants undertook were gathered using self-report measures. It is recognized that self-report data may not be reliable. It is likely that when investigating psychological constructs, such as motivation and passion, respondents will answer in a socially desirable manner to please the researchers [7]. Even when reporting behaviour such as PA, socially desirable responding is possible.

Another issue associated with measuring all the variables using self-report questionnaires is that people tend to respond in a similar way to measures that are constructed in the same kind of format, for example, using Likert scales, in which the responses are rated and are represented by numbers going from a low to a high level of agreement. This has a tendency to inflate the correlation between measures [79]. Thus, the self-report nature of the measures in this study might have overestimated the relationship between motives and passion subscales, and the relationships of motives and passion, both independently and combined, with PA.

Another limitation is that the sample in the present study was adults in Pakistan, thus, limiting the generalizability of the results to other populations. In order to give broader generalisation of the results, a larger and more diverse population could be targeted. Since participants in the present study have similar cultural backgrounds, they might respond in a similar way. People from diverse backgrounds could provide different perspectives and opinions on the same issues, in this case motives, passion, and PA, which can produce alternative perspectives that help to address this concern.

A further limitation in this study is related to selection of the sample. We used convenience sampling to select the respondents, which might limit the degree to which the results are representative of the whole Pakistani population. This is because there are socio-cultural variations between cities and rural areas, and affluent and less well-supported regions, for example, which could be reflected both in the opportunities to participate in various activities, and in attitudes to leisure-time PA. Despite these limitations, the present study is largely consistent with previous research, which suggests that there are only small differences in the types of PA people do based on cultural factors [2,23,61,80]. Nonetheless, future evaluations of demographic characteristics, including age, gender, and socio-economic status of samples from different cultures who participate in PALMS and/or passion studies, are required to confirm the present findings. In particular, there is a need for cross-cultural studies.

In the present study, there were solid psychometric findings, relating to internal consistency and the proposed models of participation motives and passion. Some studies have not included test-retest reliability because it is difficult to ask participants to repeat the measures again weeks or months after their initial measurement [37]. Without a demonstration of test-retest reliability of measures, it is difficult to draw conclusions regarding the stability of scales over time, which reduces the potential to confidently predict future behaviour, thoughts, or feelings [81,82]. In the present study, test-retest reliability was strong, both for the PALMS and for the PS, in a sizable sub-sample over a substantial period of four months.

Finally, the data collected in this study are cross sectional, so they do not permit inferences to be made about causality. Further, the analyses, primarily correlation and regression, are based on examination of association between measures, so it is possible to identify relationships, and measure the strength of associations, but associations between motives and passion, on one hand, and PA, on the other, do not reflect the conclusion that motives or passion produced differences in the amount of PA observed in this study. One way to examine causality is to conduct longitudinal research, showing that changes in one variable on the first occasion are associated with predicted changes in another variable on the second occasion [83]. Stronger evidence might also be produced by using modelling analyses, such as SEM [84]. Perhaps the most convincing method to reveal causal effects is through intervention studies [85,86]. Research in which key motives, such as mastery and enjoyment, or the key passion variable, HP, or motives and HP in combination, are increased by carefully designed interventions, and changes in PA are measured between pre- and post-intervention, should effectively determine whether motives and passion affect PA.

### 4.5. Directions for Future Research

Examination of motives for participation in PA, using measures such as PALMS, is still developing, as is research on HP and OP. Hence, there is great potential for further research on the relationship between motives and passion. Studies examining the way in which motives and passion, both independently and in interaction, affect amount of PA participation represents an important direction for research, given the need to increase PA participation globally for its recognised physical and mental health benefits. Future studies may use other research designs, such as longitudinal designs, to examine changes in motives for participation over time, or intervention studies that permit conclusions to be drawn about causality.

The theoretical basis for research on motives for participation in PA lies largely in SDT and its sub-theories [4,25], particularly organismic integration theory and cognitive evaluation theory. Systematic research using measures of motives, such as PALMS, and examining predictions of these theoretical components of SDT should be considered as a priority in this field. One issue at the heart of SDT is the psychological needs proposed to underlie motives. Deci and Ryan [24] proposed three psychological needs, namely the needs for competence, autonomy, and relatedness. The PA participation motive for mastery measures individuals’ desire to be competent at an activity [32]. The motive for enjoyment is related to the need for autonomy, because people enjoy activities when they have free choice, and do not experience a sense of being controlled [32]. The motive for affiliation is associated with the need for relatedness, as individuals search for relatedness through membership of groups and teams [32]. Research should focus on examining whether these three motives have a stronger influence on intensity and amount of PA that people do than other extrinsic motives. In the present study, one of those motives, mastery, which is an intrinsic motive, based on the need to experience competence, showed the strongest influence on participation. This is consistent with the proposition made in SDT that intrinsic motivation has the strongest long-term influence on behavior, such as PA [4]. The significant impact of HP on participation in PA that was observed in the present study is also consistent with SDT, based on the claim that passion lies at the intrinsic motivation extreme of the organismic integration continuum and that it is an intense form of intrinsic motivation [41]. HP is identified as a form of passion that is associated with autonomy, which is a key psychological need underlying self-determination in SDT. This further supports the close association between motives and HP for participation in PA.

Further research should employ more focused research designs and sensitive statistical analyses to explore the issue of whether, and in what ways, motives and passion for participation work together to influence PA participation. An obvious option for this type of research is SEM in which the relationship between motives and PA can be tested in various formats, such as models related to intrinsic versus extrinsic motives, different theoretically derived combinations of motives and passion, or more empirically, focusing on motives and passion that have shown strong associations with PA in previous research. More sophisticated research should follow in which the influence of motives is examined in relation to various demographic variables. Initial explorations have shown that gender, age, and type of PA are all related to variations in key motives [2]. Refining this research has great potential to increase our understanding about the influence of motives and passion for participation in PA on amounts and intensity of PA participation in the general population.

In the DMP, the application of passion to PA is predicted to occur through the different effects of HP and OP on PA [41,43]. In the present study, we supported the propositions that HP and OP are closely related to motives for participation in PA. At the same time, motives and passion have independent effects on PA. These important outcomes should be examined further in samples with different profiles to determine whether the close association between motives and passion, and their combined effects on PA participation observed in the present study represent stable propositions. Studies should systematically examine the effects of motives and passion, separately and together, on various types of PA. In addition, the proposition that motives and passion can operate together to increase PA participation should be further investigated. Again, one approach that could be highly informative is modelling, such as SEM. With stronger evidence that combining motives for participation and passion, especially HP, enhances PA, research should shift to intervention studies in which techniques, such as goal setting, imagery, and self-talk, are applied to enhance levels of key motives and passion, leading to increased involvement in PA from pre-intervention to post-intervention. This would provide clear evidence of a combined causal effect of motives and passion on PA.

### 4.6. Implications for Practice

This paper is not only suggestive for theory related to the role of participation motives and passion in influencing PA participation, but also has important implications for practice. In promoting PA programs, individuals’ motives and passion play a vital role in providing opportunities for behaviour change. Another important implication of this study is that adults frequently have diverse motives for engaging in PA that may mirror their essential needs and requirements. Although motivation is, at times, considered to be a global construct, investigating the various components of adults’ motivation to participate in PA might assist health professionals to develop and more accurately tailor useful interventions that aim to increase individuals’ participation in PA. It seems likely that combining enhancement of HP with interventions designed to strengthen key motives is likely to further increase the impact of interventions on the amount of PA individuals undertake.

Such interventions might not simply be useful in encouraging, recognizing, and promoting PA, but should also amplify adherence to PA. Increasing participation and adherence to PA have the potential to play a key role in reducing lifestyle-related illnesses and premature deaths among adults. Based on tenets of SDT, research has supported the proposition that intrinsic motivation is an important foundation for long-term involvement in PA, which has the potential to increase health and longevity. Finally, the findings of the present study have significant implications for the way exercise psychologists can help people to do more PA for its health benefits. The validated PALMS and PS can be used in the development of interventions, and further investigation by health authorities and sport and exercise psychologists in promoting the continued practice of PA. The Urdu language versions of PALMS and the PS will help healthcare providers, physical educators, health planners, and exercise psychologists in Pakistan to examine the motives for participation and passion of their clients, whose main spoken language is Urdu.

## 5. Conclusions

This study aimed to examine three propositions, based on SDT. First, we examined whether motives and passion were independent or related motivational variables, second, whether, independently, motives and passion enhanced PA participation, and, third, whether the impact of motives and HP on PA participation was greater when they were combined. Correlations indicated that HP was strongly, positively related to most motives for participation in PA, but OP was only weakly negatively related to motives for participation. Stepwise multiple-regression analysis indicated that mastery, physical condition, as well as HP, were related to PA participation, and accounted for a larger percentage of the variance in PA, when they acted together. Further research should be conducted to replicate the present results and to test propositions arising from the current study. With confirmation of the conclusions from this study, practitioners should be able to develop interventions to enhance motives and HP leading to increase in PA participation for health benefits in adults worldwide.

## Figures and Tables

**Table 1 ijerph-19-03298-t001:** Correlations among PALMS, Passion Scale and Physical Activity Subscales (*N* = 572).

Variables	1	2	3	4	5	6	7	8	9	10	11	12	13	14	15
1. Mastery	-	0.71 **	0.70 **	0.74 **	0.76 **	0.54 **	0.69 **	0.72 **	0.70 **	−0.33 **	0.27 **	0.35 **	0.40 **	−0.30 **	0.47 **
2. Enjoyment		-	0.52 **	0.53 **	0.64 **	0.52 **	0.61 **	0.57 **	0.56 **	−0.23 **	0.20 **	0.25 **	0.35 **	−0.26 **	0.38 **
3. Physical Condition			-	0.58 **	0.65 **	0.43 **	0.52 **	0.55 **	0.52 **	−0.22 **	0.19 **	0.29 **	0.33 **	−0.24 **	0.38 **
4. Psychological Condition				-	0.61 **	0.40 **	0.51 **	0.55 **	0.53 **	−0.23 **	0.20 **	0.25 **	0.34 **	−0.24 **	0.37 **
5. Appearance					-	0.48 **	0.54 **	0.61 **	0.58 **	−0.29 **	0.20 **	0.29 **	0.31 **	−0.23 **	0.37 **
6. Others’ Expectations						-	0.54 **	0.49 **	0.43 **	−0.20 **	0.17 **	0.22 **	0.26 **	−0.18 **	0.30 **
7. Affiliation							-	0.62 **	0.50 **	−0.19 **	0.14 **	0.26 **	0.33 **	−0.27 **	0.35 **
8. Competition								-	0.56 **	−0.24 **	0.24 **	0.28 **	0.29 **	−0.23 **	0.37 **
9. Harmonious Passion									-	−0.24 **	0.28 **	0.30 **	0.38 **	−0.28 **	0.45 **
10. Obsessive Passion										-	−0.27 **	−0.22 **	−0.29 **	0.10 *	−0.36 **
11. Vigorous Activity											-	0.36 **	0.23 **	−0.19 **	0.69 **
12. Moderate Activity												-	0.28 **	−0.22 **	0.65 **
13. Walking													-	−0.47 **	0.80 **
14. Sitting Time														-	−0.44 **
15. Overall Physical Activity															-

* *p* < 0.05; ** *p* < 0.01.

**Table 2 ijerph-19-03298-t002:** Multiple hierarchical regression analysis for measuring the significance of PALMS, harmonious passion and obsessive passion as predictors of overall PA (*N =* 572).

Variables		Physical Activity		
	R^2^	F	Β	95% CI
Step 1	0.23	42.54 **		
Mastery			87.70 **	[56.17, 119.23]
Physical Condition			26.86 *	[1.29, 52.42]
Appearance			−3.43	[−28.59, 21.73]
Competition			12.02	[−11.16, 35.21]
Step 2	0.26	38.84 **		
Mastery			59.82 **	[26.29, 93.34]
Physical Condition			23.65	[−1.57, 48.87]
Appearance			−7.23	[−32.07, 17.61]
Competition			7.37	[−15.65, 30.30]
Harmonious Passion			27.66 **	[15.11, 40.21]

** *p* < 0.01; * *p* < 0.05.

## Data Availability

The data presented in this study are available on request from the corresponding author.

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
