# Peer review of "Motives and Passion of Adults from Pakistan toward Physical Activity"

_ijerph, 2022, doi:10.3390/ijerph19063298_

Round 1
Reviewer 1 Report
IJERPH-1527783
Reviewer’s comments to authors
Thank you for the opportunity to review this manuscript. This cross-sectional study examined the relationship between motives and passion for participation in physical activity among adults, and whether motives and passion predicted physical activity. Although this is not a new research topic, it has mostly been studied in the Western population. There is merit to understanding the generalizability of behavioural change theories in Pakistan adults. I have several comments/suggestions for the authors to consider.
- The authors are suggested to re-visit the focus of this study. In the introduction and methods sections, they stated the objective of this study was to examine whether motives and passions predict physical activity. They conducted stepwise regressions to address this objective. However, they spent a few pages in the Results section presenting results from confirmatory factor analyses (CFA). Was this study trying to validate the motives and passion scales or test a theory or examine whether motives and passion are associated with physical activity? Also, please ensure appropriate statistical analysis is used to answer the research questions.
- The abstract did not reflect the methodology and results of the study.
- Introduction:
- The authors are suggested to re-organize the introduction to better orient the audiences on the research questions and related literature. As mentioned in the overall comments, the objectives of this study are unclear. When describing the literature, the authors were jumping between 2 concepts: Whether motivation predicts PA and what tools can be used to measure motives. More work is needed to articulate these concepts and relate them to the research questions.
- Many studies have examined the relationship between different types of motivation and physical activity. The authors are suggested to describe whether similar studies have been conducted in the same population. If yes, please justify how this study adds to the literature.
- The authors mentioned the SDT. This theory has validated tools for measuring different types of motives. What are the justifications for using the PALMS instead of the SDT instruments?
- Please describe the CFA procedures in the method section.
- The authors presented findings that were not mentioned in the method section, such as findings from CFA, regression analysis related to vigorous PA (method section only described overall PA as the dependent variable).
Author Response
|
Reviewer 1 |
|
|
Comment |
Reply |
|
The authors are suggested to re-visit the focus of this study. In the introduction and methods sections, they stated the objective of this study was to examine whether motives and passions predict physical activity. They conducted stepwise regressions to address this objective. |
We acknowledge the point made by Reviewer 1. The relationship between motives and passion, and their influence on PA participation represent the main thrust of the paper. Please see page 4, lines179-186.
|
|
However, they spent a few pages in the Results section presenting results from confirmatory factor analyses (CFA). Was this study trying to validate the motives and passion scales or test a theory or examine whether motives and passion are associated with physical activity? Break
|
We have justified the inclusion of information about the validation of the PALMS and PS in Urdu because validation of these measures in Urdu has not been published previously (page 5, lines 205-207). We agree with the erudite editors of the premiere publication in our field, the Psychology of Sport and Exercise journal, as well as other leading journals, which no longer publish papers validating established measures in different languages, giving priority to original work. However, readers of this study deserve to be informed about the validation process that preceded this study and upon which the present study largely depends, so we included a summary here. We consider this to be justified. However, we could summarise the validation process further, if the Editor and Reviewers believe that it will improve the flow of the manuscript, without distorting the validation process. In addition, we could cite the unpublished PhD thesis in which the validation process is described in full. |
|
Also, please ensure appropriate statistical analysis is used to answer the research questions. |
We acknowledge this comment by Reviewer 1. However, we consider that the statistical analyses we used (correlation and multiple regression) are appropriate to answer the primary research questions we posed. No change was made to the manuscript. |
|
The abstract did not reflect the methodology and results of the study. |
We have revised the Abstract to ensure that it is consistent with the Method and Results sections. We have added the following details about the translation and validation of PALMS and PS in the Abstract: Prior to data collection, we translated and validated the Physical Activity and Leisure Motivation Scale (PALMS) and the Passion Scale (PS) into Urdu. Both translated questionnaires showed acceptable internal consistency, test-retest reliability, and factorial validity (page 1, lines 15-18). |
|
The authors are suggested to re-organize the introduction to better orient the audiences on the research questions and related literature. As mentioned in the overall comments, the objectives of this study are unclear. When describing the literature, the authors were jumping between 2 concepts: Whether motivation predicts PA and what tools can be used to measure motives. More work is needed to articulate these concepts and relate them to the research questions. |
We note this point made by Reviewer 1. We have deleted parts of the Introduction to focus on the research questions about (i) the relationship between motives and passion, and (ii) the role of motives and passion in PA participation.
|
|
Many studies have examined the relationship between different types of motivation and physical activity. The authors are suggested to describe whether similar studies have been conducted in the same population. If yes, please justify how this study adds to the literature. |
We note the point made by Reviewer 1. To clarify our understanding, we consider that “Motives” are not the same as “motivation”. Motivation refers to a general psychological process, about which there are many different theories and models, and, consequently, a vast and diverse literature. Motives are specific constructs that underlie the reasons people give for their behaviour. Our focus is on motives, specifically related to physical activity. We consider that it would not be appropriate to provide a review of the extensive field of “motivation and physical activity”, but we did provide a review of theory and research related to motives and physical activity, the major focus of the present study. Information on motives for physical activity is on pages 3, lines 106-117). The secondary issue of whether there is previous research that is similar to our study is also addressed clearly in the Introduction. We are not aware of any similar published research in Pakistan. |
|
The authors mentioned the SDT. This theory has validated tools for measuring different types of motives. What are the justifications for using the PALMS instead of the SDT instruments? |
The major theoretically-derived tools were developed by Ryan and colleagues, creators of Self-determination Theory (Deci & Ryan, 1985). They are the Motivation for Physical Activity Measure (MPAM; Frederick & Ryan, 1993), which included three motives derived from SDT, namely, Interest/enjoyment, Competence, and Body-related motives; and the MPAM – Revised; Ryan et al., 1997), in which Body-related was divided into Fitness and Appearance, and the new motive, Social, was added to produce a 5-motive measure MPAM-R. We now explain in the introduction that because the MPAM and MPAM-R were developed from theory, they missed several motives that other measures had identified empirically (e.g., the Participation Motivation Questionnaire, PMQ; Gill et al., 1983). PALMS includes items measuring all the motives in the MPAM-R, with permission from Richard Ryan, as well as items from the PMQ, a widely-used, empirically-generated measure in sport psychology, but one that was not formally-validated, with permission from John Gross. PALMS is also based on our own empirical research and broadens the definitions of its eight motives. Thus, we propose that PALMS is consistent with SDT and with empirically-derived motives, but has been widely-validated in a number of languages, being demonstrated to be a robust measure, so is most suitable for use in the present research. We have revised the review of measurement of motives for participation in physical activity in the Introduction on page 2, lines 80-110, as well as the description of the PALMS in the Measures sub-section of the Method section on page 5, lines 288-290 to clarify the justification for use of the PALMS. We thank Reviewer 2 for this valuable comment, which we feel helped us to clarify this important issue. |
|
Please describe the CFA procedures in the method section. |
We have added the following sentence in the Method section: Confirmatory factor analysis (CFA) was conducted to test the validity of the translated PALMS and PS. See page 6, lines 285-287. We provide detailed description of the way we conducted the CFAs for PALMS and the PS in the subsections of the Results section, 3.1.3 and 3.2.3, that cover these procedures. |
|
The authors presented findings that were not mentioned in the method section, such as findings from CFA, regression analysis related to vigorous PA (method section only described overall PA as the dependent variable). |
The inclusion of vigorous PA as a dependent variable was an error. It has now been removed. Thanks to Reviewer 1 for pointing this out. We did conduct and report on CFA. See the previous comment and response, as well as page 8, lines 353-379 and page 9, line 402 – page 10, line 421. |

Reviewer 2 Report
The article is very well written. The is a very detailed explanation of the topic. Authors presents validation of two adapted scales and results of the motives and passion of adults from Pakistan toward physical activity.
Maybe it is worth adding additional aim of the study connected to the validation of two questionnaires: PALMS and PS
I recommend moving information why only four motives were included in regression analysis to the “ Multiple Hierarchical regression analysis” section - you mention about it in the discussion section.
And please include in the section 3.3.3.“Multiple Hierarchical regression analysis” the two aims (second and third) as it was in the first sentence of the 3.3.2. section.
I recommend dividing the second paragraph of the 4.5 section into two. It is very long.
There are some double spaces like in the lines: 82 , 501, 567, 580, 607
Thank you for this article
Author Response
|
Comment |
Reply |
|
The article is very well written. The is a very detailed explanation of the topic. Authors presents validation of two adapted scales and results of the motives and passion of adults from Pakistan toward physical activity. |
We thank Reviewer 2 for these kind words. |
|
Maybe it is worth adding additional aim of the study connected to the validation of two questionnaires: PALMS and PS |
We thank Reviewer 2 for this suggestion. We note that it is rather at odds with the comments of Reviewer 1 about the validation process, including the CFA. While we agree with Reviewer 1 that the validation process interferes with the flow of the main purposes of this study, as we discuss in our response to the comment of Reviewer 1, we also appreciate the support of Reviewer 2 for the validation process. At present, we have decided to retain the validation process in the form that is sufficient to provide a clear account. We are ready to reduce the length of the explanation, if so advised by the Editor and/or reviewers. |
|
I recommend moving information why only four motives were included in regression analysis to the “ Multiple Hierarchical regression analysis” section – you mention about it in the discussion section. |
We acknowledge the suggestion made by Reviewer 2. We have added to the Multiple Hierarchical Regression Analysis subsection, on page 12, lines 497-500, a comment about choosing 4 out of 8 motives to include in the regression analysis. However, we have also added a statement in the Analyses section of the Method on page 6, lines 294-298 that justifies the process of setting a correlation threshold. Further, we have retained a revised statement about the justification of the process in the Discussion section. |
|
And please include in the section 3.3.3.“Multiple Hierarchical regression analysis” the two aims (second and third) as it was in the first sentence of the 3.3.2. section. |
We have added aims 2 and 3 to the Multiple Hierarchical Regression Analysis subsection, on page 12, lines 491-494, as recommended by Reviewer 2. |
|
I recommend dividing the second paragraph of the 4.5 section into two. It is very long. |
As advised by Reviewer 2, we have divided this paragraph into two, splitting it at “Further research…” on page 16, line 697. |
|
There are some double spaces like in the lines: 82 , 501, 567, 580, 607 |
We have revised the paper to remove all double spaces, as recommended by Reviewer 2. |

Round 2
Reviewer 1 Report
Thank you for addressing my comments/questions. The revised version is clearer and more focused. I have several more comments/suggestions for the authors to consider.
Introduction:
- In the response letter, the authors described the difference between motivation and motives (Motivate is a construct of motivation). They also clarified motives as the focus of the study. Please add this information in the introduction.
- In low and middle-income countries,…with about 50% of the population (approximately 80 million individuals) having NCDs [53]” As this study focuses on physical activity, this information seems less relevant.
- “Researchers have indicated that there is a link between motivation and PA [35]. This suggests that research should be conducted to identify which motives for participation in PA have a noteworthy influence on PA.” Please describe the findings from previous research. For example, there was a number of studies examining the relationships between enjoyment (one of the motives being examined in this study) and physical activity.
- The authors provided justifications regarding the significance of the study. However, this information was not added to the introduction.
Methods:
- As validation of the instrument is not a study objective. I would suggest the authors briefly report the psychometric properties of the tool and present detailed results in a supplementary file.
Discussions:
- The discussions mainly focused on describing the results. The authors should compare their results to the existing literature. In the response letter, the authors justified that this study is significant because the proposed research questions have not been examined in the Pakistan population. Are the current findings consistent with the findings generated in the Western population?
Author Response
|
Reviewer 1 |
|
|
Comment |
Reply |
|
Introduction: In the response letter, the authors described the difference between motivation and motives (Motivate is a construct of motivation). They also clarified motives as the focus of the study. Please add this information in the introduction. |
We have added: Motivation refers to a general psychological process, about which there are many different theories and models, and, consequently, a vast and diverse literature. Motives are specific constructs that underlie the reasons people give for their behaviour. (Page 2, lines 62-64) |
|
Introduction: In low and middle-income countries,…with about 50% of the population (approximately 80 million individuals) having NCDs [53]” As this study focuses on physical activity, this information seems less relevant. |
We have deleted the sentence. |
|
Introduction: “Researchers have indicated that there is a link between motivation and PA [35]. This suggests that research should be conducted to identify which motives for participation in PA have a noteworthy influence on PA.” Please describe the findings from previous research. For example, there was a number of studies examining the relationships between enjoyment (one of the motives being examined in this study) and physical activity. |
We have added the following: Intrinsic motives, such as mastery, satisfaction, and enjoyment are the main drivers of long-term participation in PA [35]. It has been observed that people with high intrinsic motivation had increased benefits in activity and function [36]. (Page 3, lines 122-125)
|
|
Introduction: The authors provided justifications regarding the significance of the study. However, this information was not added to the introduction |
We have added the following: This research is significant because promoting PA is a global issue for health and wellbeing. Theory and early research suggest that enhancing motives and passion for PA could have a powerful influence on PA participation worldwide. (Page 4, lines 163-165) |
|
Methods: As validation of the instrument is not a study objective. I would suggest the authors briefly report the psychometric properties of the tool and present detailed results in a supplementary file. |
Thank you for the suggestion. We have summarized the psychometric properties of the instruments and presented the detailed results in a supplementary file. |
|
Discussions: The discussions mainly focused on describing the results. The authors should compare their results to the existing literature. In the response letter, the authors justified that this study is significant because the proposed research questions have not been examined in the Pakistan population. Are the current findings consistent with the findings generated in the Western population? |
We have revised the text to: There is research in other countries to support this prediction. In particular, intrinsic motives, mastery, and enjoyment have been linked with increased number of hours of PA participation per week [32,86,87]. HP has also been related to PA [44]. Studies by Abdullah et al. (2019), Molanorouzi et al. (2015), and Roy (2018) were conducted in Malaysia whereas the study by Ingledew and Markland (2008) was conducted in England. (Pages 14-15, lines 607-612) |
